# Fair4Free: Generating High-fidelity Fair Synthetic Samples using Data Free Distillation

## Abstract

This work presents Fair4Free, a novel generative model to generate synthetic fair data using data-free distillation in the latent space. Fair4Free can work on the situation when the data is private or inaccessible. In our approach, we first train a teacher model to create fair representation and then distil the knowledge to a student model (using a smaller architecture). The process of distilling the student model is data-free, i.e. the student model does not have access to the training dataset while distilling. After the distillation, we use the distilled model to generate fair synthetic samples. Our extensive experiments show that our synthetic samples outperform state-of-the-art models in all three criteria (fairness, utility and synthetic quality) with a performance increase of 5% for fairness, 8% for utility and 12% in synthetic quality for both tabular and image datasets.

## 1 Introduction

Nowadays, people rely on Artificial Intelligence-based applications to seek answers or make decisions. These AI-based models are trained with the data available in the real world. However, the available data in the real world is often full of machine or human biases (Liu et al., 2022). So, there is a possibility that those models will reflect biases when making a decision. For this reason, there is a strong need for bias mitigation models or bias-free datasets to ensure fairness within the models or datasets themselves. Furthermore, not all data is publicly accessible due to proprietary or sensitive cases, i.e. medical records. So, there is a need for a way to deal with these situations, too. Over the years, various approaches have been proposed to mitigate the bias issue from the model or datasets themselves, and researchers have categorized these techniques into three categories: Pre-processing, in-processing and post-processing techniques (Caton & Haas, 2024; Ntoutsi et al., 2020; Mehrabi et al., 2021). In the pre-processing technique, the dataset is processed to lower the correlation between the sensitive and non-sensitive attributes. However, in this process, valuable information can be lost. Post-processing techniques involve changing the output of the model in such a manner that the outcome of the model becomes fair towards demographics. These models also suffer accuracy as the model output is changed. In the in-processing techniques, the model is trained in such a way that during training time, the output of the model becomes fair. Though these approaches suffer from optimization problems (Oussidi & Elhassouny, 2018), a trade-off exists between fairness and accuracy. So, there is room for improvement.

Fair Generative Models (FGMs) are examples of in-processing bias-mitigation techniques. Over the years, different generative approaches have been proposed to tackle this issue. Variational autoencoder, Generative Adversarial Networks, and Diffusion-based models have seen outstanding performance in the tabular, text, and image domains (Jung et al., 2021; Choi et al., 2020; Wang et al., 2022). Different fairness constraints have been used to enforce the fairness quality in the generative samples, i.e., FairDisco (Liu et al., 2022) uses a distance correlation minimization method to weaken the connection between the sensitive and non-sensitive attributes. FairGAN (Li et al., 2022) uses dual discriminator-based generative adversarial networks architecture for generating fair synthetic samples. TabFairGAN (Rajabi & Garibay, 2022) generates synthetic fair data in two stages, first training a GAN to generate synthetic data, then adding a constraint on the synthetic samples to make it fair. FLDGMs (Ramachandranpillai et al., 2023) generate fair synthetic samples by generating the latent space with the help of GAN and diffusion architecture. Though these models generate high-fidelity fair synthetic samples, training these requires heavy computational resources; sometimes, these models do not converge while training, resulting in poor-quality synthetic samples.

Table 1: Comparison of existing fair models with our model over different key areas of interest: Fair Representation, Generative Models, and data-free training.

| Models | Fair Representation | Generative Models | Support Multiple Data Type | Data Free Training |
|---|---|---|---|---|
| TabFairGAN (Rajabi & Garibay, 2022) | ✗ | ✓ | ✗ | ✗ |
| Decaf (Van Breugel et al., 2021) | ✗ | ✓ | ✗ | ✗ |
| FairGAN (Li et al., 2022) | ✗ | ✓ | ✗ | ✗ |
| FairDisco (Liu et al., 2022) | ✓ | ✗ | ✓ | ✗ |
| FLDGMs (Ramachandranpillai et al., 2023) | ✓ | ✗ | ✓ | ✗ |
| Fair4Free (ours) | ✓ | ✓ | ✓ | ✓ |

Along these issues, if the training data is unavailable for some reason, we cannot use these models as the data is necessary to train these models.

Transferring knowledge from one learned model (teacher model) to another model (student model, generally smaller architecture than the teacher model) is known as knowledge distillation. Usually, the student model tries to learn by taking the output of the intermediate or last layer of the teacher model. This approach helps reduce computational resources. However, one of the downsides of this approach is that the data label needs to be known for distillation. Recent, fair distillation works (Dong et al., 2023; Zhu et al., 2024) rely on data labels, and these approaches cannot be used while distilling data representation.

So, from these motivations, in this work, we present Fair4Free, a novel generative approach for generating high-fidelity synthetic fair data, where we use knowledge distillation to distil the fair representation. The main contribution of our work is that while distilling the fair representation, we do not use any training data; we only use noise as input for the distilled model, so the approach for distillation is entirely data-free. The data-free distillation mitigates the issue of the dataset being sensitive and inaccessible. We first train a Variational Autoencoder (VAE) to learn the fair representation, then use it as a teacher model and take a smaller architecture (student model) to distil the fair representation. After distilling the fair representation, we use the trained decoder (from VAE) and student model to reconstruct high-fidelity fair synthetic samples. Using a smaller architecture also allows the possibility of deploying the model in edge devices. We do substantial experiments with both tabular and image data and show that our data-free distillation-based generative model performs better on the fairness, utility and synthetic samples than the state-of-the-art models. Table 1 shows the high-level comparison of other works with ours.

Our contributions to this work are as follows:

1. We present a Fair4Free, a novel data-free distillation-based fair generative model for generating fair synthetic samples.

2. Our distillation process works on latent space and requires no training data.

3. We show our distillation performance for both tabular and image datasets.

## 2 RELATED WORK

Research on data fairness and fair models has recently seen advancement due to their usefulness in real-life decision-making and other matters. To create a fair model, some research focuses on creating *fair representation*, Learning Fair Representation (LFR) (Zemel et al., 2013) create fair representation by turning the process into an optimization problem. Their results show better fairness gain performance in the downstream task. Optimal Fair Classifier (Zhao & Gordon, 2022) gives a lower bound in the classification settings to characterize the balance between accuracy and fairness. FFVAE (Creager et al., 2019) learns fair representation by disentangling the fair latent space for multiple sensitive attributes. Different adversarial approaches have been taken over the years to learn fair representations, i.e. NRL (Feng et al., 2019) learns the fair representation using a generator and a critic with the help of a min-max game they have designed. A fair contrastive learning approach has been proposed to create fair representation for datasets where sensitive attributes are not present (Chai & Wang, 2022). Besides creating fair representation, focuses have also been drawn into the *fair generative models* category. Various generative models such as VAEs, GANs, and Diffusion-based approaches have been proposed with different fairness constraints to

generate fair synthetic data in different modalities (tabular, image, texts). FairGAN (Li et al., 2022) learns exposure-based fairness by taking negative feedback while training. TabFairGAN (Rajabi & Garibay, 2022) generates tabular fair data by using a fairness constraint during the training process; Decaf (Van Breugel et al., 2021) also uses GANs to generate fair synthetic samples by using causal inference. FLDGMs (Ramachandranpillai et al., 2023) uses both GANs and diffusion architecture to generate fair latent space and reconstruct fair synthetic samples from them.

Another fair representation or model learning approach is to transfer the knowledge from one model to another, i.e., using *Knowledge distillation*. However, most of the work on knowledge distillation relies on data labels and intermediate output from the teacher model. Recently, Graph Neural Networks (GNNs) have been used to distil fair representation (Dong et al., 2023; Zhu et al., 2024) that require data labels.

## 3 PRELIMINARIES

In this section, we discuss the necessary background to follow the paper. First, we formulate the problem description followed by the definition of data fairness. Then, we discuss the background of knowledge distillation and generative models.

### 3.1 PROBLEM DEFINITION

Given a dataset tuple $(\{x_i, y_i, s_i\})_{i=1}^B \sim D$, where $x \in \mathcal{X}$ represents non-sensitive attributes, $y \in \mathcal{Y}$ is target variable and $s \in \mathcal{S}$ is sensitive attribute, we need to create fair generative model $\mathcal{H}$ by distilling fair representation ($z$) in a data-free distillation environment.

### 3.2 DATA FAIRNESS

Data fairness can be measured from different viewpoints, i.e., a model can be fair if it shows equal performance for all the demographics. Over the years, different approaches have been shown to create a fair model that satisfies group or individual fairness. This work focuses on creating a generative model that provides group fairness. To achieve this, a model should follow definitions 1 and 2.

**Definition 1** (Demographic Parity (DP), Barocas & Selbst (2016)). *A binary prediction model* $f : X \to \hat{Y}$, $\hat{Y} = \{0, 1\}$ *will achieve Demographic Parity (DP), iff*

$$P[f(X) = 1 \mid \mathcal{S} = 0] = P[f(X) = 1 \mid \mathcal{S} = 1]$$

*here, $\mathcal{S}$ is the sensitive attribute and {0, 1} are representing different groups.*

**Definition 2** (Equalized Odds (EO), Barocas & Selbst (2016)). *A binary prediction model* $f : X \to \hat{Y}$, $\hat{Y} = \{0, 1\}$ *will achieve Equalized Odds (EO), iff*

$$P[f(X) = 1 \mid Y = 1, \mathcal{S} = 0] = P[f(X) = 1 \mid Y = 1, \mathcal{S} = 1]$$

*here, $\mathcal{S}$ is the sensitive attribute and {0, 1} are representing different groups.*

### 3.3 KNOWLEDGE DISTILLATION AND GENERATIVE MODELS

*Knowledge distillation* works by taking a learned model and transfer the knowledge to a smaller architecture (smaller in layers and/or number of neurons) (Hinton et al., 2015). In the earliest distillation work (Hinton et al., 2015), the student model learned by taking the output label from the teacher model and the outcome of the student model using a supervised manner, also called response-based distillation. Most of the knowledge distillation work follows this approach (Li et al., 2023; Huang et al., 2022). However, it becomes challenging when we want to distil some distribution as they do not have any labels. Recently, Sikder et al. (2024a) used a combination of distillation loss and data-utility loss to distil the representation space without a label; however, that model requires training data to distil the latent space. Thus, distilling the model trained on private data will be difficult, especially since the data is inaccessible.

*Generative models* are used to learn data distribution and generate synthetic data that is not identical to the original data but follows the same distributions. Over the years, different kinds of generative

---

**Algorithm 1** Learning Fair Representation

---

**Input**: Biased $D$, penalty $\beta$
**Output**: Fair Encoder $\mathcal{E}_\phi$ and fair decoder $\mathcal{D}_\theta$
**Initialize**: $\mathcal{E}_\phi$ and $\mathcal{D}_\theta$ randomly

1: **for** each batch $(x_i, s_i)_{i=1}^B \sim D$ **do**
2:      Sample $z \sim \mathcal{E}_\phi(x_i, s_i)$
3:      Sample $x' = \mathcal{D}_\theta(z, s_i)$
4:      $\mathcal{L}_{\text{KL}} \leftarrow \sum_{i=1}^B D_{\text{KL}}(q_\phi(z \mid x_i, s_i) \parallel p(z))$
5:      $\mathcal{L}_{\text{re}} \leftarrow \sum_{i=1}^B \log(p_\theta(x_i \mid z, s_i))$
6:      $\mathcal{L}_{\text{final}} \leftarrow \mathcal{L}_{\text{KL}} - \mathcal{L}_{\text{re}} + \beta * \mathcal{V}_\phi^2(z, s)$; here, $\mathcal{V}_\phi^2(z, s)$ is from Equation 1
7:      Update $\mathcal{E}_\phi$ and $\mathcal{D}_\theta$ using gradient descent w.r.t. $\mathcal{L}_{\text{final}}$
8: **end for**

---

models have been proposed, i.e., Variational Autoencoder (VAE) (Kingma, 2013) tries to compress the data into latent space and reconstruct data, and Generative Adversarial Networks (GANs) (Goodfellow et al., 2020) comprised two architectures, Generator and Discriminator. These two architectures compete with each other, and after the training, the generator learns to generate high-quality synthetic samples. Diffusion models (Ho et al., 2020) learn the distribution by destroying the data by adding noise over time and then using a neural network architecture to de-noise the data. Though these approaches are beneficial and show promising outcomes, training these generative models is challenging as they do not always converge. Also, they are known for proning into mode-collapse problems.

# 4 FAIR GENERATIVE MODELS USING DATA-FREE DISTILLATION

This section presents our approach for generating a data-free, fair, generative model. The generation of fair synthetic samples involves training a fair teacher model that takes biased dataset $D$ and produces a fair representation, then distilling the fair representation using a student architecture (smaller architecture and not using the training data in the distillation process, thus data-free distillation) and finally reconstruct synthetic fair samples from the distilled fair-representation. While training the fair teacher model, we use the distance correlation minimization technique (Liu et al., 2022) to weaken the connection between the sensitive and non-sensitive attributes. We break down the entire distillation and synthetic sample generation process into three stages:

1. We first train a VAE that produces fair representation and acts as the teacher model for the distillation process. We take the biased data and minimize the relationship between the sensitive and non-sensitive features, so for any downstreaming task, the output will be free from the influence of sensitive features.

2. We use a student model (smaller architecture than the teacher model) to distil the fair representation. In this process, the student model does not have access to the training data; it takes noise as input, so this process is data-free.

3. Finally, we reconstruct high-fidelity fair synthetic samples using the distilled fair representation from the student model and the trained decoder of VAE from stage 1.

## 4.1 FAIR REPRESENTATION LEARNING

The first stage of our fair generative model is to learn a fair representation from biased data, $D$. We train a Variational Autoencoder (VAE) for this task and use the encoder ($\mathcal{E}_\phi$) to get the fair representation. The encoder creates representation $z = \mathcal{E}_\phi(x, s)$, here $(x, s) \in D$, $x \in \mathcal{X}$ represents non-sensitive attributes and $s \in \mathcal{S}$ represents sensitive attributes. Then the decoder reconstruct samples, $x' = \mathcal{D}_\theta(z, s)$. For learning the fair representation, along with the reconstruction loss and KL-divergence loss of VAE, distance correlation minimization loss $\mathcal{V}_\phi^2(z, s)$ and penalty $\beta \in \{0, 1, 2, ..., 9\}$ is used, which is stated in Equation 1 (Liu et al., 2022). Steps of learning fair representation can be found in Algorithm 1.

---

**Algorithm 2** Data-free distillation of Fair Representation

---

**Input**: Biased dataset $D$, Trained Fair Encoder $\mathcal{E}_\phi$
**Output**: Distilled model $\mathcal{E}'_\psi$
**Initialize**: $\mathcal{E}'_\psi$ randomly

1: **for** each batch $(x_i, s_i)_{i=1}^B \sim D$ **do**
2:      Sample $z \sim \mathcal{E}_\phi(x_i, s_i)$
3:      Sample $z' \sim \mathcal{E}'_\psi(n)$, where $n \sim \mathcal{N}(0, 1)$
4:      $\mathcal{L}_{\text{distillation}} \leftarrow \sum_{j=1}^k \mathcal{L}(z_j, z'_j) + \sum_{j=1}^k D_{\text{KL}}(q(z'_j) \,\|\, p(z'_j))$
5:      Update $\mathcal{E}'_\psi$ using gradient descent w.r.t. $\mathcal{L}_{\text{distillation}}$
6: **end for**

---

$$\mathcal{V}_\phi^2(z, s) = \int_{z \in \mathcal{Z}} \int_{s \in \mathcal{S}} | \, p_\phi(z, s) - p_\phi(z)p(s) \, |^2 \; dz \; ds. \tag{1}$$

## 4.2 DATA-FREE FAIR LATENT SPACE DISTILLATION AND SYNTHETIC DATA GENERATION

In this step, we take the trained fair encoder, $\mathcal{E}_\phi$ from step 1 and distil the knowledge of fair representation ($z$) to another architecture, $\mathcal{E}'_\psi$ (we use less number of hidden features than used in $\mathcal{E}_\phi$) by first creating the latent fair representation, $z = \mathcal{E}_\phi(x, s)$, where, $x, s \in D$. The main contribution of this work is to while distilling the latent space, $z$, to the model $\mathcal{E}'_\psi$, we do not use any training data, $(x, s)$. We feed Gaussian noise, $n \sim \mathcal{N}(0, 1)$ to the distilled model, and it produce some representation $z' = \mathcal{E}'_\psi(n)$. We use a combination of distillation loss between the distilled representation ($z'$) and fair representation ($z$) and KL-divergence loss on the output of the distilled model (Sikder et al., 2024a). The overall loss function is stated in Equation 2.

$$\mathcal{L}_{\text{distillation}}(z, z') = \underbrace{\sum_{j=1}^k \mathcal{L}(z_j, z'_j)}_{\text{distillation loss}} + \underbrace{\sum_{j=1}^k D_{\text{KL}}(q(z'_j) \,\|\, p(z'_j))}_{\text{KL-loss}} \tag{2}$$

Here, we use *L1-loss* as $\mathcal{L}$ for the distillation loss. Algorithm 2 shows the process of data-free distillation. After the distillation, we use the distilled model, $\mathcal{E}'_\psi$, and trained decoder, $\mathcal{D}_\theta$ from stage 1 to reconstruct high-fidelity fair synthetic samples, $\hat{x} = \mathcal{D}_\theta(\mathcal{E}'_\psi(n))$, $n \sim \mathcal{N}(0, 1)$.

## 5 EXPERIMENTS

In this section, we present the experimental analysis of our work. We utilize two tabular and two image datasets to evaluate how our model performs across various dataset types. We compare the result of our model with several state-of-the-art fair models, i.e., Decaf (Van Breugel et al., 2021), TabFairGAN (Rajabi & Garibay, 2022), FairDisco (Liu et al., 2022), FLDGMs (Ramachandran-pillai et al., 2023) in terms of fairness and utility. We further compare the works with Correlation Remover (Weerts et al., 2023) and Threshold Optimizer (Hardt et al., 2016), pre and post-processing techniques, respectively. We use FairX (Sikder et al., 2024b), a fairness benchmarking tool to load the dataset, train and evaluate the benchmark.

### 5.1 DATASET PREPROCESSING

We use four benchmarking datasets to train and evaluate our model. "Adult-Income" [1] and "Compas" (Angwin et al., 2016) are two widely used tabular dataset and "CelebA" (Liu et al., 2015) and "Colored-MNIST" (Jung et al., 2021) are image dataset. "Adult-Income" contains more than 48k records from US Census data containing personal information. "COMPAS" contains information about 7k inmates collected from the algorithm called *COMPAS* used by the US Justice system to

---

[1] https://archive.ics.uci.edu/dataset/2/adult

Figure 1: Generated Samples using Fair4Free, Dataset: CelebA, Sensitive_attribute: Smiling

predict the likelihood of an inmate re-offending. "CelebA" contains more than 200k facial images of celebrities with 40 attributes for each image. "Colored-MNIST" contains 60k English handwritten digit images with different colors ranging from 0-9. We use {Gender, Race} as sensitive attribute for the "Adult-Income" and "Compas" datasets. For the "CelebA" dataset, we use *Smiling* status, and for the Colored-MNIST, *color* as sensitive attributes. We follow the same setup of FairDisco (Liu et al., 2022) for the pre-processing and data split (80-20 ratio) technique. More details about hyperparameters in our model can be found in the Appendix section A.

## 5.2 EVALUATION METRICS

We evaluate the performance of the distilled fair representation and fair synthetic samples regarding fairness, utility and synthetic sample quality (only for the synthetic samples). For both fairness and utility evaluation, we run a downstreaming task (explained in section 5.3) and evaluate the performance of the synthetic samples on Accuracy, F1-Score, Recall (utility metrics) and Demographic Parity Ratio (DPR) (Weerts et al., 2023), Equalized Odds Ratio (EOR) (Weerts et al., 2023) (fairness utility). Along with utility and fairness evaluation, we also measure the synthetic data quality of the fair generative model. We use the Density and Coverage (Alaa et al., 2022) metrics to validate if the generated samples have the same distribution as the original samples.

Besides empirical evaluation, we also show the quality of synthetic samples and distilled latent space with visual evaluation. We use PCA (Bryant & Yarnold, 1995) and t-SNE (Van der Maaten & Hinton, 2008) plots to show how closely the distribution of the distilled latent space and fair latent space matches. Also, we show the synthetic samples generated for the image dataset.

## 5.3 DOWNSTREAMING TASK FOR EVALUATION

For the empirical evaluation, we set up a downstreaming task to determine the performance of the synthetic samples in terms of fairness and data utility. In the setup, we train a random-forest (Breiman, 2001) model for a supervised task using the features (sensitive ($s$) and non-sensitive ($x$) attributes) and target ($y$) based on the sensitive attributes, then evaluate in perspective of fairness and data utility. For example, for the "Adult-Income" dataset, we have {gender, race} as sensitive attributes. Hence, we train the random forest for each sensitive attribute and measure the fairness and data utility for respective $s$.

## 6 RESULTS AND DISCUSSION

We conduct extensive experiments and compare the performance of our model with six fair models. In this section, we discuss and analyze the result.

Figure 2: Generated Samples using Fair4Free, Dataset: *Colored-MNIST*, Sensitive_attribute: *Colors*

Table 2: Fair Synthetic Samples evaluation, Dataset: *Adult-Income*. Bold indicates the best result. Synthetic Utility evaluation is only for generative models (TabFairGAN, Decaf, FLDGMs)

| | Protected Attribute | Fairness Metrics | | Data Utility | | | Synthetic Utility | |
|---|---|---|---|---|---|---|---|---|
| | | DPR | EOR | ACC | Recall | F1-Score | Density | Coverage |
| TabFairGAN | Gender | 0.69 ± .01 | 0.60 ± .01 | 0.84± .01 | 0.61 ± .01 | 0.65 ± .01 | 0.006 ± .01 | 0.03± .01 |
| | Race | 0.026 ± .01 | 0.00 ± .00 | 0.84 ± .01 | 0.77 ± .01 | 0.67 ± .01 | 0.01 ± .01 | 0.02 ± .01 |
| Decaf | Gender | 0.52 ± .01 | 0.42 ± .01 | 0.75 ± .01 | 0.63 ± .01 | 0.44 ± .01 | 0.70 ± .01 | 0.571 ± .01 |
| | Race | 0.55 ± .01 | 0.46 ± .01 | 0.77 ± .01 | 0.69 ± .01 | 0.53 ± .01 | 0.58 ± .01 | 0.84 ± .01 |
| FLDGMs (DM) | Gender | 0.94 ± .01 | 0.94 ± .01 | 0.69 ± .01 | 0.90 ± .01 | 0.81 ± .01 | **1.26** ± .01 | 0.89 ± .01 |
| | Race | **0.99** ± .01 | 0.96 ± .01 | 0.69 ± .01 | 0.91 ± .01 | 0.81 ± .01 | **1.24** ± .01 | 0.86 ± .01 |
| FairDisco (base-model) | Gender | 0.98 ± .01 | 0.85 ± .01 | 0.78 ± .01 | 0.92 ± .01 | 0.86 ± .01 | n/a | n/a |
| | Race | 0.95 ± .01 | 0.92 ± .01 | 0.812 ± .01 | 0.71 ± .01 | **0.88** ± .01 | n/a | n/a |
| Correlation-Remover | Gender | 0.32 ± .01 | 0.23 ± .01 | 0.86 ± .01 | 0.65 ± .01 | 0.71 ± .01 | n/a | n/a |
| | Race | 0.29 ± .01 | 0.20 ± .01 | 0.86 ± .01 | 0.80 ± .01 | 0.71 ± .01 | n/a | n/a |
| Threshold Optimizer | Gender | 0.95 ± .01 | 0.35 ± .01 | 0.86 ± .01 | 0.66 ± .01 | 0.65 ± .01 | n/a | n/a |
| | Race | 0.69 ± .01 | 0.25 ± .01 | 0.87 ± .01 | 0.66 ± .01 | 0.71 ± .01 | n/a | n/a |
| Original Data | Gender | 0.32 ± .01 | 0.22 ± .01 | **0.88** ± .01 | 0.80 ± .01 | 0.72 ± .01 | n/a | n/a |
| | Race | 0.19 ± .01 | 0.00 ± .00 | **0.88** ± .01 | 0.80 ± .01 | 0.71 ± .01 | n/a | n/a |
| Fair4Free (ours) | Gender | **0.99** ± .01 | **0.99** ± .01 | 0.76 ± .01 | **0.98** ± .01 | **0.88** ± .01 | 1.03 ± .01 | **0.96** ± .01 |
| | Race | **0.99** ± .01 | **0.99** ± .01 | 0.76 ± .01 | **0.98** ± .01 | 0.87 ± .01 | 1.20 ± .01 | **0.97** ± .01 |

## 6.1 EMPIRICAL ANALYSIS FOR TABULAR DATA

We show the performance of our model in the downstreaming task regarding fairness and data utility for the "Adult-Income" dataset in Table 2 and the "Compas" dataset in Table 3. We use the {gender, race} as sensitive attributes and record the results for both tables. For the "Adult-Income" dataset, we predict the income class for an individual given their attributes (both sensitive and non-sensitive) as downstreaming task. And for the "Compas" dataset, we predict the re-offend probability for an inmate given their previous records.

**Data Utility Analysis**  We measure the *Accuracy, Recall* and *F1-score* from the downstreaming task. We observe from Table 2 that our synthetic samples achieve 5% and 8% better performance in fairness and utility compared to FLDGMs (state-of-the-art model). In Table 3, the performance of our samples' Synthetic utility (Coverage) is better than all other methods. We also achieve a balance of fairness and accuracy scores compared with other models.

**Synthetic Quality Analysis**  Synthetic samples should perform better for both data utility and synthetic quality. Overall, the performance of the synthetic quality of Fair4Free is better than other generative models. Such as, for Table 3, though TabFairGAN and Decaf perform better in the utility

Table 3: Fair Synthetic Samples evaluation, Dataset: *Compas*. Bold indicates the best result. Synthetic Utility evaluation is only for generative models (TabFairGAN, Decaf, FLDGMs)

| | | Fairness Metrics | | Data Utility | | | Synthetic Utility | |
|---|---|---|---|---|---|---|---|---|
| | **Protected Attribute** | DPR | EOR | ACC | Recall | F1-Score | Density | Coverage |
| TabFairGAN | Gender | $0.52 \pm .01$ | $0.42 \pm .01$ | $\mathbf{0.68} \pm .01$ | $0.63 \pm .01$ | $\mathbf{0.66} \pm .01$ | $0.016 \pm .01$ | $0.014 \pm .01$ |
| | Race | $0.50 \pm .01$ | $0.49 \pm .01$ | $\mathbf{0.69} \pm .01$ | $0.59 \pm .01$ | $\mathbf{0.64} \pm .01$ | $0.014 \pm .01$ | $0.011 \pm .01$ |
| Decaf | Gender | $0.87 \pm .01$ | $0.84 \pm .01$ | $0.45 \pm .01$ | $0.40 \pm .01$ | $0.42 \pm .01$ | $0.367 \pm .01$ | $0.39 \pm .01$ |
| | Race | $\mathbf{0.99} \pm .01$ | $\mathbf{0.96} \pm .01$ | $0.45 \pm .01$ | $0.40 \pm .01$ | $0.42 \pm .01$ | $0.38 \pm .01$ | $0.39 \pm .01$ |
| FLDGMs (DM) | Gender | $0.92 \pm .01$ | $0.91 \pm .01$ | $0.52 \pm .01$ | $0.41 \pm .01$ | $0.44 \pm .01$ | $\mathbf{1.40} \pm .01$ | $0.134 \pm .01$ |
| | Race | $0.98 \pm .01$ | $0.86 \pm .01$ | $0.53 \pm .01$ | $0.45 \pm .01$ | $0.47 \pm .01$ | $\mathbf{1.59} \pm .01$ | $0.13 \pm .01$ |
| FairDisco | Gender | $0.97 \pm .01$ | $0.92 \pm .01$ | $0.55 \pm .01$ | $0.40 \pm .01$ | $0.43 \pm .01$ | n/a | n/a |
| | Race | $0.87 \pm .01$ | $0.76 \pm .01$ | $0.53 \pm .01$ | $0.41 \pm .01$ | $0.44 \pm .01$ | n/a | n/a |
| Correlation-Remover | Gender | $0.43 \pm .01$ | $0.33 \pm .01$ | $0.64 \pm .01$ | $0.64 \pm .01$ | $0.59 \pm .01$ | n/a | n/a |
| | Race | $0.58 \pm .01$ | $0.63 \pm .01$ | $0.65 \pm .01$ | $\mathbf{0.64} \pm .01$ | $0.60 \pm .01$ | n/a | n/a |
| Threshold Optimizer | Gender | $0.92 \pm .01$ | $\mathbf{0.98} \pm .01$ | $0.65 \pm .01$ | $\mathbf{0.65} \pm .01$ | $0.61 \pm .01$ | n/a | n/a |
| | Race | $\mathbf{0.99} \pm .01$ | $0.76 \pm .01$ | $0.63 \pm .01$ | $0.63 \pm .01$ | $0.60 \pm .01$ | n/a | n/a |
| Original Data | Gender | $0.37 \pm .01$ | $0.28 \pm .01$ | $0.57 \pm .01$ | $0.65 \pm .01$ | $0.61 \pm .01$ | n/a | n/a |
| | Race | $0.54 \pm .01$ | $0.58 \pm .01$ | $0.66 \pm .01$ | $0.57 \pm .01$ | $0.61 \pm .01$ | n/a | n/a |
| Fair4Free (ours) | Gender | $\mathbf{0.99} \pm .01$ | $0.95 \pm .01$ | $0.52 \pm .01$ | $0.41 \pm .01$ | $0.42 \pm .01$ | $1.12 \pm .01$ | $\mathbf{0.97} \pm .01$ |
| | Race | $0.93 \pm .01$ | $0.84 \pm .01$ | $0.52 \pm .01$ | $0.40 \pm .01$ | $0.42 \pm .01$ | $1.06 \pm .01$ | $\mathbf{0.98} \pm .01$ |

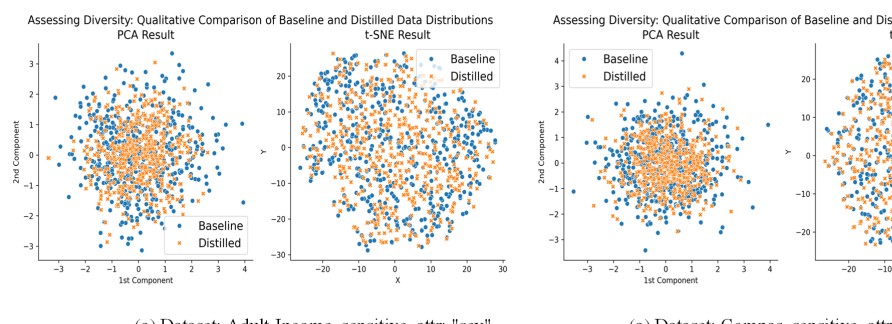

(a) Dataset: Adult-Income, sensitive_attr: "sex"          (a) Dataset: Compas, sensitive_attr: "sex"

Figure 3: PCA and t-SNE plots of the distilled and original fair representation for both *Adult-Income* (left) and *Compas* (right) Dataset. If the orange (distilled) and blue (original) dots overlap, it signifies their distribution is similar.

metrics, their synthetic utility performance is worse than our scores. We achieve 85% better scores in the Coverage metrics than the Decaf (Compas dataset) and 12% better performance than the FLDGMs in the *Adult-Income* dataset.

## 6.2 VISUAL ANALYSIS FOR TABULAR AND IMAGE DATA

Besides the empirical quality of our synthetic samples, we also show visual analysis. We show the synthetic image samples from both *CelebA* and *Colored-MNIST* in Figure 1 and 2. We use "Smiling" as the sensitive attribute for the *CelebA* and "Colors (Red, Green, Blue)" for the *MNIST* dataset.

For the tabular dataset, we use the PCA and t-SNE plots to show how closely the distribution matches the distilled representation and the original fair representation. We observe from Figure 3 that the distributions closely match each other; this signifies that our data-free distillation method transfers the knowledge correctly.

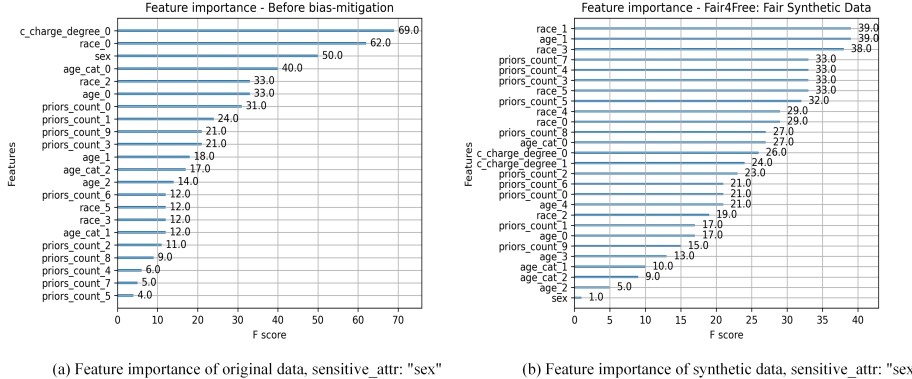

(a) Feature importance of original data, sensitive_attr: "sex"  (b) Feature importance of synthetic data, sensitive_attr: "sex"

Figure 4: Feature Importance Analysis with the original (left) and synthetic data, Dataset: *Compas*, Sensitive Attribute: *Sex*

### 6.3 FEATURE IMPORTANCE ANALYSIS

Along with the empirical evaluation and visual evaluation, we also test the feature importance of the original and synthetic data in downstreaming task. We set up a task, i.e., for the *Compas* dataset, given the inmate's records, to predict how likely the person will re-offend. Figure 4 shows the feature importance for both original and synthetic data (in this case, we use *Gender (sex)* as a sensitive attribute.). We observe that, for the original dataset, gender plays a vital role in reaching the decision, on the contrary, our synthetic samples do not rely on the sensitive attribute for making a decision. This also proves the usefulness of our synthetic samples for fair decision-making process.

### 6.4 DISCUSSION

This work presents Fair4Free, a data-free distillation-based fair generative model. With this approach, we generate high-fidelity fair synthetic samples with the help of knowledge distillation. We distil the fair representation from the trained model (teacher model) to another architecture (student model). During the distillation process, we do not use any training data; thus, the training of the student model is data-free. This helps when the dataset is unavailable for security and privacy reasons. Also, as we use a small architecture, we reduce the computational cost, and we can deploy the model into an edge device with better performance than the teacher model. Tables 2 and 3 show that our model is over-performing the state-of-the-art models in the perspective of data utility, fairness and synthetic quality. Besides empirical evaluation, Figure 1, 2 shows the synthetic data samples, and Figure 4 shows the usefulness of fair synthetic samples in decision-making.

**Social Impact**  The Fair Generative model can play a vital role in the decision-making process because the generated samples do not consider the sensitive attributes while making decisions. This process helps to reduce bias and discrimination in the real-world scenario, i.e. fair recommendations in the healthcare system, fair economic recommendation (decision making on loans). Also, the fair model can help build trust with the user so it can be widely used in society.

**Limitations and Future Works**  In our experiments, we use a single sensitive attribute to train the generative model, i.e., we use either Gender or Race (for both Adult-Income and Compas datasets). So, in order to tackle intersectional bias, we need to work on a generative model that handles multiple sensitive attributes. However, we believe the data-free distillation process we present in this work can also be used if we have fair representation with multiple sensitive attributes.

## 7 CONCLUSION

Real-world data is filled with human and/or machine biases, and in the era of AI-powered decision-making systems, these biased data can cause harm to specific people as these models are trained with

them. Furthermore, some datasets are not available publicly due to proprietary cases or restricted by data protection laws like GDPR or HIPPA to protect privacy. So, one way to tackle the data limitation and bias issue is to use a generative model. This work presents Fair4Free, a novel fair generative model that uses data-free distillation to generate fair synthetic samples. We pre-train a VAE with the biased data and produce fair representation in the latent space, then use another architecture to distill the fair representation. The distillation process is completely data-free, and then we use the distilled fair representation to create fair synthetic samples. Our extensive experiment shows that the quality of the synthetic samples outperforms state-of-the-art models regarding fairness, utility, and synthetic quality. As we use a distillation process and smaller architecture for the distilled model, these models can be deployed in the edge devices. Also, as the synthetic samples are fair towards demographics, these can help to mitigate the biased data issue.

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

## A  MORE EXPERIMENTAL DETAILS

**Hyperparameters**  To train the benchmarking models, we use FairX (Sikder et al., 2024b), a fair benchmarking tool. FairX uses the same hyperparameters for the respective benchmarking models, specified in their original publication. For our generative model, we follow the same architecture of FairDisco (Liu et al., 2022) for the VAE setup (teacher model) and Table 4 shows the hyperparameters for the distillation model (student model).

Table 4: Hyperparameters for the Distillation Model of Fair4Free (Tabular Dataset)

|  | **Parameters** |
| --- | --- |
| Linear Layer 1 | $64(noise\_dim) \rightarrow 32$ |
| Linear Layer 2 | $32 \rightarrow 2 \times 8(hidden\_dim)$ |
| Batch Size | 2048 |
| Epochs | 5000 |
| Learning rate | $1e-5$ |
| Optimizer | $Adam$ |

**Workstation Setup**  We train our model and run the benchmark using the same environment that is equipped with "AMD Ryzen 9 5900x 12-core processor, 128 GB RAM, NVIDIA GeForce RTX 3090 Ti with 24 GB of GPU memory".

