# OpenReview forum: "Fair4Free: Generating High-fidelity Fair Synthetic Samples using Data-Free Distillation"
_ICLR.cc/2025/Conference — Submitted to ICLR 2025_

### Official Review · Reviewer_Z6so · 2024-10-15

**Soundness:** 2
**Presentation:** 2
**Contribution:** 2
**Rating:** 3
**Confidence:** 2

**Summary:**

This work proposes a generative model for fair synthetic data generation with data-free distillation. The proposed method involves first training a fair teacher model on the biased dataset and producing a fair representation, then distil using a student architecture. Finally, synthetic fair samples are reconstructed from the distilled fair representation. The authors apply the method on image and tabular data, and outperform all baseline methods.

**Strengths:**

1. The empirical results look promising as they outperform most baselines.
2. The experiment is performed on both visual and tabular data
3. The proposed method is well presented and easy to understand

**Weaknesses:**

1. The teacher model trained in this paper is assumed to have fair representation. This involves using a regularized distance correlation minimization loss to weaken the connection between the sensitive and non-sensitive attributes. This limits the applicability of the proposed method (i.e. distilling from pre-trained large generative models without a fairness guarantee).
2. Experiment on visual data (i.e. CelebA and Colored-MNIST) has no quantitative benchmarking and comparison with baseline methods. Comparison with baseline such as EDM [1] and showing better fairness can strength the soundness of the experiment, even without outperforming it in terms of visual quality.
3. The claim about data-free training is a bit confusing; please refer to the Questions section.

[1] Karras, T., Aittala, M., Aila, T. and Laine, S., 2022. Elucidating the design space of diffusion-based generative models. Advances in neural information processing systems, 35, pp.26565-26577.

**Questions:**

I'm a bit confused about the motivation for using data-free distillation. The authors claim that the method's benefit is that it can be applied when data is inaccessible; however, the teacher model required in the method is trained with a designed loss function to have a fair representation.

I assume the underlying assumption of this work is that the existing foundational generative models do not have the desired fairness property, so in reality, the original training data is still required to apply the proposed method. Then, my question is, why does data-free distillation matter in this case? Would it be easier to use a single model to learn how to generate fair data from scratch?

As a non-practitioner in fairness, I would appreciate if the author could clarify if I misunderstood the setting or the motivation of the work.

**Details Of Ethics Concerns:**

This work aims to generate synthetic data to mitigate human bias in real-world data so that the model can be more fair.

---

> ### Author Response · Authors · 2024-11-25
> **Rebuttal Comments**
>
> Thank you for the comments. Here are some clarrifications:
>
> 1. Imagine the following scenario:
>
>      Some entity wants to release its trained model to a third party but cannot share the data itself, and the teacher model (encoder) requires training data to perform its task. So, if the third party wants to transfer the knowledge of the teacher model to another model (smaller in size preferably), one option is to use knowledge distillation. So, in this case, our presented approach in this paper, data-free distillation can solve this task, which does not require any dataset.
>
> So, the main contribution of our work is data-free distillation in latent space. We opened the possibility to distil fair model in case the data is unavailable. So, we use the distillation process to transfer the knowledge to a different model.
>
> 2. We have added some quantitative scores for the Image dataset, we used Density and Coverage metrics to show the synthetic quality of the generated image samples (higher the better results). We also calculated FID score. We compare our work with FLDGMs only as other generative models in our study were originally created for tabular datasets. And we will compare our work with the suggested paper by the reviewer in future.
>
> | Models                         | Dataset       | Density | Coverage | FID  |
> |--------------------------------|---------------|---------|----------|------|
> | FLDGMs                         | CelebA        | 1.29    | 0.89     | 1.49 |
> |                                | Colored-MNIST | 1.19    | 0.814    | 3.93 |
> | Data-Free Distillations (Ours) | CelebA        | 0.89    | 0.94     | 1.63 |
> |                                | Colored-MNIST | 1.62    | 0.78     | 3.85 |

---

> > ### Comment · Reviewer_Z6so · 2024-11-25
> >
> > Thank you to the authors for their response. However, I still believe the quality of the work does not convince me to lean toward acceptance. Further clarification regarding the motivation and contributions combined with proper ablation study is needed to justify the method design. In the rebuttal, I understand the general privacy-critical setting, but I am unclear about how it connects to fairness. Specifically, if you have a fair teacher model (i.e. learning an unbiased distribution), wouldn't distilling from it generally result in a fair student model?

---

### Official Review · Reviewer_emFZ · 2024-11-01

**Soundness:** 2
**Presentation:** 2
**Contribution:** 2
**Rating:** 3
**Confidence:** 3

**Summary:**

Fair4Free presents a novel approach to generating synthetic data that is fair and unbiased using data-free distillation. This method is particularly useful when access to training data is restricted due to privacy concerns. The process involves distilling knowledge from a pre-trained teacher model to a smaller student model without using real data, relying instead on noise input. The paper claims significant improvements over other models in terms of fairness, utility, and synthetic quality.

**Strengths:**

1.  The data-free approach allows the generation of fair data even when access to original datasets is restricted, addressing significant privacy concerns.
2.  By using a smaller student model, Fair4Free reduces computational resources and enables potential deployment on edge devices.
3.  The model demonstrates superior performance across fairness, utility, and synthetic quality metrics compared to state-of-the-art models.

**Weaknesses:**

1. Complexity and Overhead: While reducing computational costs via a smaller model, the initial setup of training and distilling between models can be complex and resource-intensive.
2.  How does the performance of synthetic samples generated by Fair4Free compare to non-synthetic data in terms of real-world utility and fairness?
3. How robust is Fair4Free to shifts in data distributions that might occur in practical scenarios?

**Questions:**

see weaknesses.

---

> ### Author Response · Authors · 2024-11-25
> **Rebuttal Comment**
>
> Thank you for the comments. Here are some clarrifications:
>
> 1. Regarding the comment of complexity and overhead, we agree with the reviewer, but the amortized lose will be lower as a consequence because of the smaller student model.
>
> 2. We tested our model on widely accepted benchmarking datasets in the fairness field. Table 2 and 3 reports the Fairness, Utility of both generated samples and original data (non-synthetic) in down streaming task (details in section 5.3 and Section 6.1).  It’s our expectation that as these are real-world datasets, our model will also work on other diverse datasets.
>
> 3. The datasets we used in this work (Adult-Income and COMPAS) are real world datasets. COMPAS dataset was created from the output of an algorithm used by the US justice system and the Adult-Income dataset is taken from the US census. Our model did capture the data distributions of these datasets which proves the robustness of our model. Furthermore, we can improve the robustness by using architectural modifications or using loss functions improvement [1].
>
> References:
>
> [1] Feng, Yeli, Daniel Jun Xian Ng, and Arvind Easwaran. "Improving variational autoencoder based out-of-distribution detection for embedded real-time applications." ACM Transactions on Embedded Computing Systems (TECS) 20.5s (2021): 1-26.

---

### Official Review · Reviewer_9nCi · 2024-11-01

**Soundness:** 3
**Presentation:** 2
**Contribution:** 2
**Rating:** 3
**Confidence:** 3

**Summary:**

Fair4Free introduces an innovative generative model designed to create fair synthetic data without accessing actual datasets. It leverages a technique called data-free distillation, where knowledge is transferred from a teacher model to a smaller student model using only noise as input. The approach is touted for its effectiveness in generating data that adheres to fairness, utility, and quality benchmarks, surpassing existing models.

**Strengths:**

1. Enhanced Privacy Protection: The model's ability to operate without real data makes it highly suitable for environments with strict privacy regulations or sensitive data restrictions.
2.  By employing a smaller model architecture, Fair4Free minimizes the computational demands, making it feasible for deployment on less powerful devices, including edge devices.
3. It consistently outperforms other models in generating synthetic data that scores highly on fairness, utility, and quality metrics, as validated by rigorous experimental evaluations.

**Weaknesses:**

1.  The model currently focuses on addressing bias with respect to single sensitive attributes, potentially overlooking complex bias scenarios involving multiple intersecting attributes.
2. Scalability Concerns: While the model is efficient, scaling it to handle larger or more complex datasets without compromising performance remains a challenge.
3. Can Fair4Free be adapted to efficiently manage multiple sensitive attributes to tackle intersectional biases more effectively?
4. In terms of decision-making and predictive accuracy, how do the synthetic datasets generated by Fair4Free compare to those derived from traditional data generation methods?
5. Adaptability to Data Shifts: What measures can be taken to enhance Fair4Free's robustness against dynamic changes in data distribution that are common in real-world settings?

**Questions:**

See weaknesses.

---

> ### Author Response · Authors · 2024-11-25
> **Rebuttal comment**
>
> Thank you for the comments. Here are some clarrifications:
>
> 1. We believe the data-free distillation process we present in this work can also be used if we have fair representation with multiple sensitive attributes. We need to modify our teacher model in a way that can handle multiple sensitive attributes. But this is out of the scope of this paper, we mentioned this in the limitation sections of the paper.
>
> 2. We are aware of the situation on the handling larger of more complex dataset, but from the base-line perspective, we used 4 benchmarking real-world datasets, where the CelebA (image) dataset contains more than 200K face images, and this is one of the largest image datasets. Also, we used two data domains (tabular and image). Our model performs well in these datasets.
>
> 3. If we have a teacher model that can handle multiple-sensitive attributes, the same approach of data-free distillation of Fair4Free can be used to transfer the knowledge of multiple sensitive attributes. Again, handling the multiple-sensitive attributes is out of the scope of this paper.
>
> 4. Table 2 and 3 reports the Fairness and Utility of both generated samples and original data in down streaming tasks (details in section 5.3 and Section 6.1). We compare our work with different traditional generative methods (GANs, Diffusion). Assuming the reviewer meant GANs or Diffusion model as traditional data generation methods, otherwise we would appreciate a clarification.
>
> 5. The datasets we used in this work (Adult-Income and COMPAS) are real world datasets. COMPAS dataset was created from the output of an algorithm used by the US justice system and the Adult-Income dataset is taken from the US census. Our model did capture the data distributions of these datasets which proves the robustness of our model. Furthermore, we can improve the robustness by using architectural modifications or using loss functions improvement [1].
>
> References:
>
> [1] Feng, Yeli, Daniel Jun Xian Ng, and Arvind Easwaran. "Improving variational autoencoder based out-of-distribution detection for embedded real-time applications." ACM Transactions on Embedded Computing Systems (TECS) 20.5s (2021): 1-26.

---

### Official Review · Reviewer_LGEY · 2024-11-04

**Soundness:** 1
**Presentation:** 1
**Contribution:** 1
**Rating:** 3
**Confidence:** 4

**Summary:**

The authors propose Fair4Free, a novel generative approach for generating high-fidelity synthetic fair data using knowledge distillation.
Their proposed method consists of three stages: 1) train a VAE on the biased dataset to learn a fair representation, 2) use it as a teacher model and distill the fair representation to a student model, and 3) use the trained VAE decoder and student model to reconstruct high-fidelity fair synthetic samples. They show experiment results on tabular and image datasets.

**Strengths:**

* The paper addresses an important problem of generating synthetic fair data.
* The experiments include a wide range of evaluation metrics to thoroughly evaluate the proposed method.

**Weaknesses:**

* The contributions of this paper are unclear. Although the authors claim the method is *data-free*, it still relies on a biased dataset to train the teacher model in the first stage. Furthermore, the method combines the student encoder with the teacher decoder to generate synthetic samples. Then why not simply use the teacher model directly? What advantages are gained by distilling the teacher encoder into a smaller student encoder only to recombine it with the teacher decoder? The paper also lacks an ablation study to justify the design choices in the method.
* Overall, the paper needs substantial improvement in writing quality and clarity. Implementation details are severely lacking, e.g., hyperparameters used for training the proposed and compared methods, details on synthetic data generation during evaluation, how the random forest model is trained, and explanation of the evaluation metrics (e.g., DPR, EOR).
* The method demonstrates minimal performance gains over the compared methods across downstream tasks (Table 2,3).
* They should additionally report FID for synthetic data quality.

**Questions:**

* The paper needs to improve the clarity of implementation details and evaluation protocols mentioned above.
* The paper needs an ablation study to justify the design choices of the method.

---

> ### Author Response · Authors · 2024-11-25
> **Rebuttal comment**
>
> Thank you for the comments. Here are some clarrifications:
>
> 1. Imagine the following scenario:
>
>      Some entity wants to release its trained model to a third party but cannot share the data itself, and the teacher model (encoder) requires training data to perform its task. So, if the third party wants to transfer the knowledge of the teacher model to another model (smaller in size preferably), one option is to use knowledge distillation. So, in this case, our presented approach in this paper, data-free distillation can solve this task, which does not require any dataset.
>
> So, the main contribution of our work is data-free distillation in latent space. We opened the possibility to distil fair model in case the data is unavailable. So, we use the distillation process to transfer the knowledge to a different model
>
> 2. Our work's implementation details (data pre-processing techniques and hyperparameters) are mentioned in section 5.1 and Appendix A. For training the benchmarks, we use FairX [1], which uses the original hyperparameters of the respective publications. We use off-the shelf Random Forest from Sklearn to run the downstreaming task. And we are adding the explanation for EOR and DPR. Thank you for your suggestions.
>
> 3. We evaluated our model with state-of-the-art methods in terms of Fairness, Data-Utility and Synthetic utility and we gain a performance improvement of  5%, 8% and 12% performance respectively, also, we achieve a score of 0.99 out of 1.00 for the Demographic parity ratio for both datasets (Table 2, 3), we achieve better performance than the teacher model due to the usage of a combination of KL-loss and distillation loss. So, we disagree with the reviewer that the performance improvement is minimal.
>
> 4. We use **Density** and **Coverage** metrics for evaluating the synthetic quality of our generated samples. These methods are widely accepted for evaluating synthetic samples. Our work is based on both tabular and image dataset, the suggested FID score gives synthetic quality only for Image datasets. Was the reviewer suggesting FID score only for Image dataset? Maybe the reviewer can give us some pointer where original FID score was used for tabular dataset, so we can use them in our work. However, we have added FID evaluation for the synthetic images for our model. We compare the results with the FLDGMs as other generative models in our study were originally developed for the tabular datasets.
>
> | Models                         | Dataset       | FID Score |
> |--------------------------------|---------------|-----------|
> | FLDGMs                         | CelebA        | 1.49      |
> |                                | Colored-MNIST | 3.93      |
> | Data-Free Distillations (Ours) | CelebA        | 1.63      |
> |                                | Colored-MNIST | 3.85      |
>
> References:
>
> [1]. Sikder, Md Fahim, et al. "FairX: A comprehensive benchmarking tool for model analysis using fairness, utility, and explainability." AEQUITAS, co-located with ECAI 2024, 2024.

---

> > ### Comment · Reviewer_LGEY · 2024-12-03
> >
> > Thank you for your response. However, I still have remaining concerns:
> >
> > * The provided scenario does not fully address my question: Then why not simply use the **teacher model** directly? What **advantages** are gained by distilling the teacher encoder into a smaller student encoder only to recombine it with the teacher decoder? If the teacher model already achieves a fair representation, it’s unclear what additional benefit or novelty is introduced by this distillation process.
> > * Is the benefit **better performance** over the teacher model? Due to lack of clarity in the main text, it is unclear what is the "teacher model" baseline in Table 2 and 3.

---

> > > ### Author Response · Authors · 2024-12-04
> > > **Comment**
> > >
> > > 1. **Response**: Teacher model (encoder part) requires training data during training process and inference process, and if the data is inaccessible, the teacher model cannot be used! Whereas if you distill the encoder part into a smaller network in data-free distillation process like ours, then:
> > > 	1. you don't need training data to train or infer from the distilled encoder
> > > 	2. Its smaller in size, hence take less memory to train (one can argue, training the teacher itself is resource intensive, thats true, but the amortized lose will be lower as a consequence because of the smaller student model)
> > > 	3. As we showed in our evaluation that distilled encoder can generate the same fair representation as the teacher encoder, so, you can use the teacher decoder to reconstruct dataset. It is not possible to use both teacher encoder and decoder if the training data is inaccessible.
> > > 	4. We did not distill the teacher decoder as our distilled encoder follows the same distribution as the teacher encoder so we can reuse the teacher decoder. So, our approach is more sustainable in terms of resource usage!
> > > 2. **Response**: Its just not the "better performance" over the teacher model. As we mentioned in the response 1, data-free distillation allow us to distill the fair representation without the access of the training data! And regarding the teacher model clarification, Section 4.1 of the paper should be helpful to understand it.

---

### Meta-Review · Area_Chair_uEch · 2024-12-08

**Metareview:**

The submission proposes "Fair4Free," a generative approach to creating synthetic fair data via data-free distillation. While the paper addresses an important issue in fairness and privacy, it fails to convincingly justify its contributions and practical advantages. Below is a summary of the key feedback:

Strengths:
* Addresses a relevant problem of fairness in synthetic data generation.
* The approach operates in privacy-restricted scenarios, which is a valuable consideration.
* Includes evaluation across multiple metrics on tabular and image datasets.

Weaknesses:
* Unclear Contributions: The method's claim of being "data-free" is misleading as it still relies on a biased dataset to train the teacher model. Reviewers questioned the necessity of distilling knowledge from the teacher model rather than using it directly. The novelty and advantage of the distillation process, particularly when reusing the teacher model’s decoder, are insufficiently justified.
* Limited Empirical Validation: Quantitative results demonstrate marginal improvements over baselines, and the performance gains (5%-12%) are not compelling. Key experimental details, such as hyperparameters and ablation studies, are missing, undermining the reproducibility and validation of design choices. Evaluation on image datasets (e.g., CelebA, Colored-MNIST) lacks robust benchmarking against advanced baselines (e.g., diffusion models).
* Presentation and Clarity: The writing lacks clarity, and the methodology is not well-explained. Several questions from reviewers (e.g., on teacher-student architecture rationale and multi-attribute fairness) remain inadequately addressed.
* Scalability and Applicability: The approach struggles with scaling to larger datasets or complex bias scenarios, limiting its generalizability.
Intersections of multiple sensitive attributes, an essential aspect of fairness, are not addressed.

Based on the unclear novelty, lack of robust validation, and insufficient responses to reviewer concerns, I recommend rejection. While the paper has potential, it requires significant improvements in methodology, experimental rigor, and presentation to meet the standards of acceptance.

**Additional Comments On Reviewer Discussion:**

Reviewer Points:
* LGEY: Questioned the necessity of distillation over direct use of the teacher model; criticized lack of clarity and detail.
* 9nCi: Highlighted limitations in handling multiple attributes, scalability, and robustness to data shifts.
* emFZ: Concerned about complexity and overhead, comparison with non-synthetic data.
* Z6so: Clarification needed on motivation for data-free distillation in fairness context; criticized the assumption of fairness in the teacher model.

Author Responses:
* LGEY: Explained that distillation enables data-free operation, reducing data dependency and resource use.
* 9nCi: Acknowledged multi-attribute handling as out of scope, justified model performance on existing datasets.
* emFZ: Clarified complexity overhead vs. long-term resource benefits, pointed to reported comparisons with real data.
* Z6so: Clarified the privacy scenario for distillation but did not fully resolve the fairness motivation.

---

### Decision · Program_Chairs · 2025-01-22

Reject